# Analysis of logistics capacity, influencing factors and spatial spillover effect in Yangtze River Economic Belt

**Li Fanghu[1], He Yinnan 📷[1]\*, Wang Biao[2]**

1 School of Economics and Management, Huainan Normal University, Huainan, Anhui, China, 2 College of Shipping, Ningbo University, Ningbo, Zhejiang, China

\* yinluofengyin@163.com

## Abstract

The logistics industry plays a crucial role in facilitating regional economic development. Serving as a vital link connecting producers, consumers, and various components of the supply chain, it has a direct and profound impact on the prosperity and advancement of regional economies. Based on the panel data of 11 provinces in the Yangtze River Economic Belt from 2016 to 2020, this paper constructs the logistics capability evaluation index system from four aspects: regional economic base, logistics infrastructure, logistics development scale, information technology and talent support, and uses the entropy weight TOPSIS method to measure the logistics capability of each province. The adjacency space weight matrix, geographical distance weight matrix and economic distance weight matrix are selected to build a spatial econometric model to analyze the influencing factors and spatial spillover effects of regional logistics capability. The following conclusions can be drawn from the analysis. From 2016 to 2020, the regional logistics capacity of the Yangtze River Economic Belt shows a trend of increasing year by year, but the logistics capacity of different provinces within the region has a large room for improvement. From the perspective of spatial dimension, the logistics capacity of the Yangtze River Economic Belt is "high in the east and low in the west". The results of spatial econometric analysis based on the spatial Durbin model show that there are significant spatial spillover effects on the logistics capacity of provinces in the Yangtze River Economic Belt. Factors such as road network density, port throughput, water freight turnover, transportation, warehousing and postal employment will not only affect the logistics capacity of the region, but also have a spillover effect on the material capacity of neighboring provinces in the Yangtze River Economic Belt. This study improves the level of regional logistics capacity and promotes the regional economic development of the Yangtze River Economic Belt. It can be used as a reference for other regions or countries in terms of enhancing regional logistics capacity and promoting regional economic development.

**Data Availability Statement:** All files are available from the Years of statistics in China database (http://data.stats.gov.cn/index.htm).

**Funding:** This work was supported by the scientific research project of Anhui province (NO. 2023AH051548) and the Anhui Province quality

engineering teaching research project (NO. 2022jyxm1437) The leader of the scientific research project of Anhui province(NO. 2023AH051548) is Yinnan He. the leader of the Anhui Province quality engineering teaching research project (NO. 2022jyxm1437) is Fanghu Li. The authors' contributions are as follows: Conceptualization: Fanghu Li, Yinnan He, Biao Wang. Data curation: Fanghu Li, Biao Wang. Formal analysis: Fanghu Li, Yinnan He, Biao Wang. Funding acquisition: Fanghu Li, Yinnan He. Methodology: Fanghu Li, Yinnan He, Biao Wang. Supervision: Fanghu Li, Yinnan He. Validation: Fanghu Li, Yinnan He, Biao Wang. Writing – original draft: Fanghu Li, Biao Wang. Writing – review & editing: Fanghu Li, Yinnan He, Biao Wang.

**Competing interests:** The authors have declared that no competing interests exist.

## Introduction

The logistics industry has a significant correlation effect that plays an important role in promoting industrial structure adjustment and economic development. At present, the Yangtze River Economic Belt is China's fastest growing regional economy. The Chinese government has taken many policy measures to promote regional economic development. As the largest regional economic complex in China, the Yangtze River Economic Belt plays an important role in promoting the level of logistics capacity in the process of regional economic development. In 2014, the Chinese government released the Yangtze River Economic Belt Comprehensive Three-Dimensional Transportation Corridor Plan (2014–2020), specifying the layout plan of the Yangtze River mainline river crossings [1]. In 2020, the "Yangtze River Main Line River Crossing Layout Plan (2020–2035)" issued by China's National Development and Reform Commission set higher requirements for the coordinated development of the Yangtze River waterway and for the ecological environmental protection, flood control and shipping safety [2]. It is very important to understand the status of the regional logistics capacity and its main influencing factors, as such understanding will be conducive to the high-quality development of the regional economy.

Studying the logistics capacity and spatial layout of the Yangtze River Economic Belt plays an important role in improving the regional comprehensive strength and competitiveness and is also an important measure to promote the rapid and stable development of the regional economy. The development of the regional logistics industry can have a correlation and interaction effect with other industries, can constantly promote the adjustment and upgrading of the regional industrial structure and the high-quality development of the economy, and can play an exemplary and promoting role in the whole country. Therefore, the coordinated development of the logistics industry in the Yangtze River Economic Belt is not only an important part of regional coordinated development but also an objective need for the development of the modern logistics industry. The Yangtze River Economic Belt runs across Eastern China, Central China, and Western China, covering the coastal areas in the east and the vast inland areas. The logistics capacity of different regions is different. The evaluation of the logistics capacity of the Yangtze River Economic Belt can lead to a comprehensive understanding of the regional logistics capacity current state of various provinces and cities, can lead to grasping the relationship between the unbalanced development of regional logistics capacity and interregional logistics capacity, and can provide a theoretical basis for promoting the overall development of regional logistics.

Research on logistics ability begins with enterprise logistics ability. Enterprise logistics capability is a comprehensive concept that encompasses multiple aspects. From a transportation perspective, a company's logistics capability is the sum of the design parameters of transportation equipment and is an important asset of the company. From the perspective of customer service, logistics capability refers to the provision of competitive customer service at the lowest possible cost [3]. Logistics capability is a measure of a company's logistics performance in important aspects such as cost, quality, delivery, flexibility, and innovation. Enterprises can use and can allocate resources reasonably to achieve maximum efficiency and improving customer care needs [4]. Logistics capability includes the capacity of logistics facilities, equipment, or systems, as well as the logistics operation capability of enterprises, as this is the comprehensive level of logistics functions. For example, Bowersox et al. argue that logistics capability is a relative assessment of a manufacturer's ability to provide competitive customer service at the lowest possible total cost [5]. Daughtery et al. explored logistics capabilities directly on a time-based basis and identified delivery flexibility as an important logistics capability element [6]. Fawcett notes that an important aspect to improve the international operation performance of

enterprises is the logistics capability based on time, including delivery capability, quality capability and resilience capability [7].

At the enterprise level, logistics capability is mainly reflected in personalized and customized services, use by carriers, delivery accuracy, customer feedback, application of advanced transportation tracking systems, cycle time consistency, delivery lead time, delivery frequency and delivery punctuality, load rate and order completion rate. Enterprise logistics capability is a key factor affecting enterprise performance and supports the realization of enterprise strategy [8]. The logistics capability of enterprises can be divided into two categories: logistics association capability and logistics operation capability. Logistics correlation ability is mainly reflected in customer response ability and communication ability between enterprises and customers. Logistics operation capability is reflected in delivery reliability, delivery speed, product availability and product quality [9]. There are many types of industries studying logistics capability, such as the consumer goods industry, e-commerce industry, fast food industry, automobile manufacturing industry and so on.

With the deepening of the research, the research scope of logistics capability continues to expand, from the enterprise level to the supply chain level and the regional logistics level. Regarding supply chain logistics capability, scholars emphasize the role of information technology in the supply chain. Information technology is an important basis for enterprises to implement supply chain integration. Information technology could manage information flow and is the bridge of communication and cooperation between all parties in the supply chain [10]. The logistics information technology capabilities of enterprises can promote information sharing among supply chain partners and significantly improve supply chain capabilities. For example, Wu et al. conducted a survey on 184 manufacturing enterprises and concluded that the IT progress of enterprises could significantly improve their supply chain capabilities [11]. Li et al. investigated 182 Chinese manufacturing enterprises and concluded that the effective implementation of information technology plays an important role in supply chain integration [12].

Lynch studied the decision-making problem of the logistics facility location. He emphasized the necessity of strengthening coordination among multinational enterprises and proposed that the production and distribution network system is an important tool to build the global supply chain of enterprises [13]. Jorg believes that the transformation of enterprise organization and merger and reorganization will lead to the redesign and layout of the regional logistics system [14]. Therefore, he proposed that the regional logistics system should be divided into three layers of logistics infrastructure, transportation network platform and material demand plan for unified planning to avoid the redesign and waste in the logistics system caused by the change in the enterprise cooperation mode. Lu established a comprehensive regional logistics hub model and applied mixed integer linear programming, greedy heuristic method, particle swarm optimization algorithm and Hungarian method to determine the location of logistics facilities at different levels of the supply chain and determine the demand for expansion or closure of facilities in different time periods [15]. Guo used an MLP neural network to predict regional logistics demand [16]. By constructing the LMDI model, panel quantile model and dependent variable limited model, Cheng et al. analyzed the development mode and influence effect of the regional logistics industry in China and explored the main ways and promotion paths to promote the development of the logistics industry [17]. Zhang et al. explicitly consider the interaction between logistics management departments and logistics users, as well as the impact of economies of scale and carbon dioxide emission taxes on the design of logistics networks [18].

Regional logistics capability refers to the ability of regional logistics system to organize, integrate and utilize various resources within the system in a certain period of time to fully meet customer needs. Regional logistics capability can be divided into broad regional logistics

capability and narrow regional logistics capability. The generalized regional logistics capability refers to the logistics system capability that can be composed of multiple countries and regions with common development characteristics and development needs. Regional logistics in the narrow sense refers to the logistics system composed of similar regional scope within a country.

In the study of regional logistics capability, the evaluation of regional logistics capability is an important content. The steps of regional logistics capability evaluation are as follows: first, identify the object of evaluation; Secondly, we decompose the influencing factors, construct the evaluation index system, and test whether it is reasonable. Then, the evaluation model is selected, including index weight model and comprehensive evaluation model. Finally, the evaluation model is applied to a concrete example to analyze the evaluation results.

Wenfeng Zhao establishes a regional logistics capability evaluation system by using analytic Hierarchy Process (AHP). This paper analyzes and evaluates the development of regional logistics in 6 typical provinces, and puts forward some suggestions to improve regional logistics capability and promote the healthy development of regional logistics [19]. Carlucci et al. used principal component analysis to explore the development level of regional logistics in northeast Italy, and evaluated the regional logistics competitiveness based on this [20]. Xiaoling Huang et al. took the above seaport and Busan port as examples, and used the three-stage data envelopment analysis method to evaluate the efficiency of the two ports respectively [21]. Peter et al. conducted an in-depth study on the synergistic development of regional logistics agglomeration and economic agglomeration in the United States from the perspective of regulating correlation variables and improving accuracy, and realized the synergistic effect of regional logistics and regional economy [22]. Sun Ting analyzed the development level of regional logistics in Guangxi by constructing a first-level index system from four aspects: economic foundation, logistics infrastructure construction, regional logistics demand capacity, logistics information and talent competitiveness [23]. Hang Jiang used entropy weight TOPSIS method and FSQCA method to evaluate low-carbon logistics by using panel data of 30 provinces in China from 2008 to 2021 [24]. Zhang Guangsheng selected 10 indicators from three levels: logistics demand capacity, logistics facilities and economic capacity, and logistics development potential to build a comprehensive evaluation system for regional logistics capacity [25]. He Xiaoguang selected 14 indicators from four aspects: economic development level, logistics supply and demand level, logistics innovation level and logistics industry scale to construct regional logistics indicators, and analyzed the impact of regional logistics capability on enterprise performance by using regression analysis [26]. Based on the new development concept, Shi Peizhe selected 19 evaluation indicators from seven aspects: economic development, logistics operation, innovation, coordination, green, development and sharing, and used fuzzy mature-element method and coefficient of variation method to measure the regional logistics capacity of Henan Province [27].

Logistics capability evaluation is a complicated system problem, which involves many factors. It can be seen from the existing research literature on regional logistics capacity that the evaluation indicators mainly focus on regional economic development and logistics infrastructure, and the summary of other influencing factors on regional logistics capacity is incomplete. For example, the lack of consideration of logistics capacity output, logistics informatization and human resources factors.

The research methods mainly include analytic hierarchy process, principal component analysis, grey correlation analysis, factor analysis, regression analysis and cluster analysis, etc. Spatial econometrics method is rarely used in the research of regional logistics capability.

From the perspective of research topics, the current research content is more inclined to the measurement of logistics capacity in a certain field, such as the measurement of green

logistics capacity, regional agricultural product logistics capacity, low carbon logistics capacity, etc. There are few studies on the regional logistics capacity of the Yangtze River Economic Belt.

As a typical representative of China's regional economic development, the Yangtze River Economic Belt plays an exemplary role in China's regional economic development, and provides reference for other countries to promote regional economic development. Logistics plays an important role in promoting regional economic development. There are few researches on the regional logistics capacity of the Yangtze River Economic Belt. Therefore, this paper intends to take the regional material capacity of the Yangtze River Economic Belt as the research object, explore the temporal and spatial evolution characteristics of the regional logistics capacity of the Yangtze River Economic Belt, and analyze the influencing factors and spatial spillover effects of the logistics capacity. Specifically, the main contribution of this paper includes two aspects. First, the comprehensive index system of regional logistics capacity evaluation is constructed from four aspects: regional economic development level, logistics infrastructure, logistics business volume, human resources and informatization level, and the development level of regional material capacity of the Yangtze River Economic Belt is systematically evaluated, as well as the spatio-temporal change characteristics. Second, considering that spatial effect analysis rarely appeared in previous studies, this paper adopts appropriate spatial econometrics methods to analyze the influencing factors and spatial spillover effects of regional logistics in the Yangtze River Economic Belt, providing certain empirical basis for formulating differentiated strategies for improving regional logistics capacity to narrow the regional logistics gap and promote regional coordinated development.

The rest of this paper is organized as follows: The second part mainly introduces the research methods used in this paper. The third part introduces the construction of index system and data collection. The fourth part of the Yangtze River logistics capacity measurement and space-time characteristics analysis. The fifth part is space measurement, analysis and policy suggestion. The sixth part is Conclusion.

## Methods

### Entropy weight TOPSIS methods

The concept of TOPSIS (Technique for Order Preference by Similarities to Ideal Solution) is a technique for solving multiple criteria decision-making problems. TOPSIS is suitable for cases with many attributes, alternatives and is handy for objectives with quantitative data because it alleviates the requirement of paired comparisons, and the capacity limitation may not significantly dominate the process. However, weight attributes are of great importance in TOPSIS. It is necessary to develop weighting algorithm calculations to maintain the consistency of judgment of the decision-making method. For the sake of the objectivity of the evaluation, the entropy weighting method was adopted to avoid subjective effects. The solution technique comprises a series of stages as follows [28].

Step 1 Create and standardize the analysis matrix.

Determine the elements of the matrix according to the established evaluation system so that $y_{ij}$ is the i-th indicator of the evaluation object j.

$$Y = (y_{ij})_{m \times n} = \begin{pmatrix} y_{11} & \cdots & y_{1n} \\ \vdots & \ddots & \vdots \\ y_{m1} & \cdots & y_{mn} \end{pmatrix} (i = 1, 2, \cdots, m; j = 1, 2, \cdots n)$$

Step 2: Construct the normalized matrix.

Normalized matrix Z is as follows:

$$Z = (z_j)_{m \times n} = \begin{pmatrix} z_{11} & \cdots & z_{1n} \\ \vdots & \ddots & \vdots \\ z_{m1} & \cdots & z_{mn} \end{pmatrix}$$

Where,

$$z_{ij} = \frac{y_{ij}}{\sum_{i=1}^{m} y_{ij}} \ (j = 1, 2, \cdots n) \tag{1}$$

Step 3: Calculate the entropy of each attribute $H(x_j)$.

$$H(x_j) = -k \sum_{i=1}^{m} Z_{ij} \ln Z_{ij} \ (j = 1, 2, \cdots, n) \tag{2}$$

Where, $k = \frac{1}{\ln m}$

Step 4: Determine weight.

The normalized weighted coefficient $w_{ij}$ is as follows:

$$w_j = \frac{1 - H(x_j)}{n - \sum_{j=1}^{m} H(x_j)} \ (j = 1, 2, \cdots, n) \tag{3}$$

Where, $0 \leq w_j \leq 1, \sum_{j=1}^{m} w_j = 1$.

Step 5: The entropy weight evaluation matrix is constructed

The weighted evaluation matrix X

$$X = \begin{bmatrix} x_{11} & \cdots & x_{1n} \\ \vdots & \ddots & \vdots \\ x_{m1} & \cdots & x_{mn} \end{bmatrix} = \begin{bmatrix} z_{11}w_1 & \cdots & z_{1n}w_n \\ \vdots & \ddots & \vdots \\ z_{m1}w_1 & \cdots & z_{mn}w_n \end{bmatrix}$$

Step 6: Determine the positive ideal and negative ideal solutions.

Positive ideal solution:

$$x^+ = \{x_1^+, x_2^+, \cdots, x_m^+\}$$

$$= \{(\max(x_{\theta ij}|j \in J_1), \min(x_{\theta ij}|j \in J_2)|1 \leq \theta \leq y, 1 \leq i \leq n)\} \tag{4}$$

Negative ideal solution:

$$x^- = \{x_1^-, x_2^-, \cdots, x_m^-\}$$

$$= \{(\min(x_{\theta ij}|j \in J_1), \max(x_{\theta ij}|j \in J_2)|1 \leq \theta \leq y, 1 \leq i \leq n)\} \tag{5}$$

Where, $x^+$ is the set of positive ideal solutions, $x^-$ is the set of negative ideal solutions, $J_1$ represents the set of positive attributes, $J_2$ represents the set of negative attributes.

Step 7: Calculate the distance measures for each alternative.

The distance from the positive ideal alternative is

$$D_i^+ = \sqrt{\sum_{\theta=1}^{y} \sum_{j=1}^{m} (w_j(x_{\theta ij} - x^+)^2)}, i = 1, 2, \cdots, n \tag{6}$$

Similarly, the distance from the negative ideal alternative is:

$$D_i^- = \sqrt{\sum_{\theta=1}^{y} \sum_{j=1}^{m} (w_j(x_{\theta ij} - x^-)^2)}, i = 1, 2, \cdots, n \qquad (7)$$

Step 8: calculate the relative closeness to the ideal solution $C_i^*$

$$C_i^* = \frac{D_i^-}{D_i^+ + D_i^-}, i = 1, 2, \cdots n, C_i^* \in [0, 1] \qquad (8)$$

## Spatial autocorrelation model

Spatial autocorrelation is mainly used to describe the correlation between regional spatial units. In this paper, global Moran's I index and local Moran's I index are used to represent the spatial relationship of the logistics capacity [29], and the calculation formula is as follows:

Calculating global Moran's I

$$I = \frac{m}{W_0} \times \frac{\sum_{i=1}^{m} \sum_{j=1}^{m} w_{ij}(y_i - \bar{y})(y_j - \bar{y})}{\sum_{i=1}^{m}(y_i - \bar{y})^2}, W_0 = \sum_{i=1}^{m} \sum_{j=1}^{m} w_{ij} \qquad (9)$$

Where, m is the number of space units, $y_i$ and $y_j$ are the i-th and j-th index values, $\bar{y}$ is the average value of the index; $w_{ij}$ is the spatial weight matrix.

The local Moran's $I_i$ index is used to reflect the spatial agglomeration of local areas, and its calculation formula is as follows:

$$I_i = \frac{y_i - \bar{y}}{S^2} \sum_{i \neq j}^{m} w_{ij}(y_i - \bar{y}), S^2 = \frac{1}{n} \sum_{i=1}^{m}(y_i - \bar{y}) \qquad (10)$$

## Spatial econometric model

In order to verify the spatial spillover effects of the logistics capacity of the Yangtze River Economic Belt and its influencing factors among provinces, this paper introduces spatial measurement models. The more commonly used spatial econometric models include spatial lag model (SLM), spatial error model (SEM) and spatial Durbin model (SDM). Spatial Durbin model can be converted into spatial lag model and spatial error model under certain conditions [30]. Since the purpose of this paper is to analyze the spatial spillover effect of regional logistics capacity and its influencing factors, this paper mainly focuses on whether it is possible to choose the spatial Durbin model that contains endogenous and exogenous interaction effects. The model is set as follows:

$$y_{it} = \alpha + \rho \sum_{j=1, j \neq i}^{n} W_{ij} y_{jt} + Z_{it} \varphi + \theta \sum_{j=1}^{n} W_{ij} Z_{ij} + m_{it} \qquad (11)$$

$$m_{it} = \lambda \sum_{j=1, j \neq i}^{n} W_{ij} m_{it} + \varepsilon_{it} \qquad (12)$$

Where: $m_{it}$ is the random error term, obeying an independent homogeneous distribution; $W_{ij}$ is the spatial matrix of n×n; $Z_{it}$ is the explanatory variable; ρ is the spatial autoregressive coefficient, which represents the endogenous interaction effect, i.e., the spatial spillover effect of the explanatory variable; θ is the parameter to be estimated, which represents the exogenous interaction effect, reflecting the spatial spillover effect of the explanatory variable; and λ is the spatial autocorrelation coefficient, which represents the interaction effect between the error terms.

## Selection of the spatial weighting matrix

The traditional measurement model only studies the influence of individual independent variables on individual dependent variables, while the spatial measurement model further explores the influence of individual independent variables on other individual dependent variables on the basis of traditional measurement. Therefore, the greatest feature of spatial metrology is to fully consider the spatial correlation between cross-sectional units. Spatial correlation refers to the spatial interdependence, mutual restriction, mutual influence and interaction between things and phenomena in different regions. It is the inherent spatial economic attribute of things and phenomena, and the essential attribute of geographical spatial phenomena and spatial processes. The spatial correlation is caused by the proximity of geographical location, the competition and cooperation of individuals at the cross-section level, the group imitation behavior, the spillover effect of economic activities and the measurement error. Therefore, in the actual construction of the model, how to incorporate the spatial effects of economic variables into the model, it is necessary to establish a spatial weight matrix according to some standards. To define the spatial weight, we must first quantify the position of the spatial unit, which usually includes neighborhood space distance, geographical distance, economic distance and industrial distance.

In this paper, we constructed the neighborhood space weight matrix, geographic distance weight matrix and economic distance weight matrix for spatial measurement analysis.

Neighborhood space weight matrix. According to the spatial adjacency, using the principle of Queen connection, the element $W_{ij}$ in the matrix of $W_{0-1}$ takes the value of 1 when the regions i and j are neighboring, and $W_{ij}$ takes the value of 0 when the regions i and j are not neighboring.

Geographic distance weight matrix. It is assumed that the spatial effect depends on the distance between spatial units, and the closer the distance between spatial units, the stronger the spatial effect, which is $W_{dis}$ The matrix $1/d_{ij}$ is the inverse of the geographic distance between 2 provinces in the same region, which is calculated by latitude and longitude.

$$W_{dis} = \begin{cases} 1/d_{ij}, & i \neq j \\ 0, & i = j \end{cases} \tag{13}$$

Economic distance weight matrix. The smaller the gap in the level of economic development between regional provinces, the closer the systems and infrastructure between provinces, and the more similar the level of logistics capacity, thus constructing the economic distance weight matrix $W_{eco}$, whose element $W_{ij}$ is the absolute value of the inverse of the difference in per capita GDP between two provinces in the same region, and $e_i$ is the per capita GDP of the province i.

$$W_{eco} = \begin{cases} 1/|e_i - e_j|, & i \neq j \\ 0, & i = j \end{cases} \tag{14}$$

## Index system and data

From the perspective of connotation and characteristics, combined with existing research findings, this paper constructs a logistics capability evaluation indicator system from four aspects: regional economic foundation, logistics infrastructure, logistics development scale, information technology, and talent support, as shown in Table 1.

(1) Regional Economic Foundation

**Table 1. Evaluation index of regional logistics capability.**

| Elements layer | Indicator layer | Symbol | Unit |
|---|---|---|---|
| Regional economic foundation | GDP | V1 | 100 million yuan |
| | Total retail sales of consumer goods | V2 | 100 million yuan |
| | Total import and export value | V3 | 100 million yuan |
| Logistics infrastructure | Total road mileage | V4 | Km |
| | Road network density | V5 | Km/Km^2 |
| | Cargo truck ownership | V6 | ten thousand vehicles |
| Logistics development scale | Goods turnover volume | V7 | hundred million ton-kilometers |
| | Total freight volume | V8 | hundred million tons |
| | Production value of transportation, storage, and postal services | V9 | 100 million yuan |
| | Port throughput | V10 | hundred million tons |
| | Road freight | V11 | ten thousand tons |
| | Total revenue of express delivery services | V12 | 100 million yuan |
| | Water transport freight turnover | V13 | hundred million ton-kilometers |
| Information technology and talent support | Total volume of postal and telecommunication services | V14 | 100 million yuan |
| | Length of optical cable lines | V15 | Km |
| | Technology market turnover | V16 | 100 million yuan |
| | Number of Internet users | V17 | people |
| | Number of college students per 100,000 population | V18 | people |
| | Number of research and development personnel | V19 | people |
| | Employment in transportation, storage, and postal industries | V20 | ten thousand people |

The regional economic foundation and logistics capability support each other. The regional economy determines the scale of regional logistics demand, while logistics supports the development of the regional economy, promoting the flow of goods, capital, and technology. This paper selects three indicators to reflect the regional economic foundation: provincial GDP, total retail sales of consumer goods, and total import and export value [24, 31].

GDP is a measure of the economic situation within a territorial area, indicating the total value of all final products and services produced in an economy within a certain period (usually a year). It is widely recognized as the best indicator for measuring a country or region's economic condition.

Total Retail Sales of Consumer Goods refers to the total amount of physical goods sold to individuals and social groups by enterprises (units) within a region for non-production and non-business use, as well as the income from catering services. It reflects the vitality of the consumer goods market.

Total Import and Export Value is an important indicator of a country's or region's economic performance in international trade. It reflects the trade of goods and services between that country or region and other countries. The magnitude of a region's total import and export value can reflect the strength of its logistics capability.

(2) Logistics Infrastructure

Logistics infrastructure is an important carrier for regional logistics activities and a significant support for the development of the logistics industry. Well-developed logistics infrastructure and convenient transportation networks are essential for meeting the logistics development needs of the Yangtze River Economic Belt and enhancing the comprehensive logistics capability of the entire region. Accordingly, this paper selects three indicators to evaluate the regional logistics infrastructure: total road mileage, road network density, and cargo truck ownership [26, 32].

Total Road Mileage is a key indicator reflecting the scale of road construction development and an important symbol of socio-economic development levels. It is also a crucial foundation for the development of the logistics industry. This paper chooses the total road mileage as one of the indicators for regional logistics infrastructure.

Road Network Density refers to the network structure formed by roads of different functions, levels, and locations within an urban area. The road network density equals the total length of all roads in a certain calculation area divided by the total area of the region.

Cargo Truck Ownership indicates the number of trucks used for transporting goods in a region, country, or specific economic system. It is one of the important indicators of the level of economic activity in a country or region. Trucks, as one of the main tools of logistics transportation, directly affect the efficiency of the supply chain. A sufficient number of trucks can meet the circulation needs of goods more timely and flexibly, contributing to the efficiency of the entire logistics system.

(3) Logistics Development Scale

The scale of logistics operations refers to the overall scale and level of logistics activities in a region, involving aspects such as logistics freight volume, goods turnover, goods value, etc. The development of this scale has profound impacts on the economy, society, and environment. This paper selects indicators like goods turnover volume, total freight volume, production value of transportation, storage and postal services, port throughput, road freight volume, total revenue of express delivery services, and water transport freight turnover to reflect the scale of logistics operations [25, 33].

Goods Turnover Volume refers to the sum of the product of the quantity of goods and their corresponding transportation distances over a certain period. It is an important indicator reflecting the total output of the transportation industry, and also crucial for calculating labor productivity, transportation efficiency, and unit transportation costs.

Total Freight Volume represents the total amount of goods actually transported by various means of transportation over a period, indicating the service level of the transportation industry for the people and the national economy, as well as an important indicator for studying the development scale and speed of the transportation industry.

Production Value of Transportation, Storage, and Postal Services is an important indicator in regional economic statistics, used to measure the value created by these industries in the economy. Transportation, storage, and postal services are vital components of the logistics industry, and their development directly affects the operation and growth of other industries such as manufacturing, retail, and services.

Port Throughput refers to the total amount of goods loaded and unloaded at ports and handled over a certain period. This indicator is key to assessing a port's operational condition and economic vitality. High throughput typically indicates busy trade activities and robust economic growth, as more goods are needed for import and export.

Road Freight Volume measures the total amount of goods transported through the road transport system, indicating the scale and level of goods transportation activities in a country or region. High freight volume means that the transportation network is functioning well, allowing goods to move quickly and efficiently from production sites to consumer markets, also signifying economic activity growth and business prosperity.

Total Revenue of Express Delivery Services is directly related to e-commerce activities. The express delivery business is a crucial part of the e-commerce supply chain, and an increase in its revenue reflects the development of e-commerce and the supply chain. With economic growth and enhanced commodity mobility, people's demand for express delivery services also increases, driving the growth of express delivery business revenue, making it an important component of logistics operations.

Water Transport Freight Turnover mainly reflects the capacity of regional waterway transportation, including not only the quantity of transport objects but also the factor of transportation distance, thus comprehensively reflecting the transportation production results. It is a primary basis for calculating transportation efficiency, labor productivity, and transportation unit costs.

(4) Information Technology and Talent Support

The development of information technology provides technical support for the development of enterprises, enhancing their innovation and competitive advantages. The role of human resources in regional or enterprise development is not just supportive but strategic, directly affecting the innovation level and long-term competitiveness of the region or enterprise. Information technology and human resources form a powerful support system through collaborative cooperation. This paper selects indicators like total postal and telecommunications services, length of optical cable lines, technology market turnover, number of internet users, number of college students per 100,000 population, number of professional and technical personnel per 10,000 population, and employment in transportation, storage, and postal industries to reflect the status of regional information technology and human resources [27, 34].

Total Volume of Postal and Telecommunication Services usually includes the total of postal and telecommunications services. With the advent of new technologies, changes in social demands, and the development of the global communications industry, this indicator varies, reflecting the total development results of postal and telecommunications services over a certain period. It is a comprehensive indicator to observe the overall trend of postal and telecommunications business development and reflects the level of logistics informatization in a region.

Length of Optical Cable Lines refers to the actual length of long-distance optical cables used for transmitting optical signals. In the information age, the length of optical cable lines reflects the level of regional information and communication technology and the status of informatization construction, and is also an important indicator of regional logistics informatization.

Technology Market Turnover refers to the total amount of transactions of various technology products, services, hardware, and software in the field of science and technology over a specific time. This concept covers a wide range of technology fields, including but not limited to computer science, information technology, biotechnology, engineering technology, etc. The turnover of the technology market can reflect the overall market size, industry growth trends, industry innovation, and technological trends.

Number of Internet Users refers to the number of independent individuals using the internet over a certain period. This includes people accessing the internet using computers, smartphones, tablets, or other devices. The number of internet users is one of the important indicators to measure the popularity and influence of the internet. It not only reflects the technical popularity of the internet but also has profound effects on various aspects such as business, government, and society.

Number of College Students per 100,000 Population reflects the educational level of personnel in a region, representing the country's emphasis on talent cultivation. Simultaneously, technical talents will make significant contributions to the development of various industries.

Number of Research and Development Personnel indicates the number of people engaged in scientific research and technological innovation activities over a certain period. These individuals are dedicated to advancing new technologies, products, or services, contributing to social and economic progress. The number of research and development personnel plays a crucial role in promoting social, economic, and technological development.

Employment in Transportation, Storage, and Postal Industries reflects the labor market conditions of these industries, playing an important role in meeting daily production needs

and promoting economic development. It is an important indicator of the regional logistics capability system.

This paper's analysis covers 11 provinces in China (including Shanghai, Jiangsu, Zhejiang, Anhui, Jiangxi, Hubei, Hunan, Chongqing, Sichuan, Guizhou, and Yunnan). Related data of all variables, spanning 2016 to 2020, were sourced from the China Statistical Yearbook [35].

## Regional logistics capacity calculation and spatial autocorrelation test

### The result of entropy weight TOPSIS method

Step 1 Data collection

The statistical yearbooks of provinces and cities in the Yangtze River Economic Belt from 2016 to 2020 were searched, and relevant data were obtained according to the indicators in Table 1.

Step 2 Calculate index weight

The data collected from 2016 to 2020 are standardized according to Formula 1. According to Formulas 2 and 3, the entropy and weight of each index are calculated, and the average value of each year's index weight is taken as the final index weight, as shown in Table 2.

Step 3 Computational logistics capability using the TOPSIS method

The TOPSIS method was applied to determine the level of logistics capacity of the provinces and cities, and the results are shown in Table 3 and Fig 1.

A comprehensive overview of the development of logistics capabilities in various provinces and cities along the Yangtze River Economic Belt (Table 2 and Fig 1) reveals significant disparities. High logistics capability values are mainly concentrated in the downstream Yangtze River region, specifically the Yangtze River Delta area, while low values are concentrated in the

**Table 2. Index weight results.**

| Elements layer | Indicator layer | Symbol | Weight | Total weight |
|---|---|---|---|---|
| Regional economic foundation | GDP | V1 | 0.0409 | 0.1609 |
| | Total retail sales of consumer goods | V2 | 0.0329 | |
| | Total import and export value | V3 | 0.0871 | |
| Logistics infrastructure | Total road mileage | V4 | 0.0188 | 0.0732 |
| | Road network density | V5 | 0.0301 | |
| | Cargo truck ownership | V6 | 0.0243 | |
| Logistics development scale | Goods turnover volume | V7 | 0.0713 | 0.4591 |
| | Total freight volume | V8 | 0.0411 | |
| | Production value of transportation, storage, and postal services | V9 | 0.0221 | |
| | Port throughput | V10 | 0.0991 | |
| | Road freight | V11 | 0.0135 | |
| | Total revenue of express delivery services | V12 | 0.0962 | |
| | Water transport freight turnover | V13 | 0.1160 | |
| Information technology and talent support | Total volume of postal and telecommunication services | V14 | 0.0524 | 0.3068 |
| | Length of optical cable lines | V15 | 0.0461 | |
| | Technology market turnover | V16 | 0.0588 | |
| | Number of Internet users | V17 | 0.0437 | |
| | Number of college students per 100,000 population | V18 | 0.0212 | |
| | Number of research and development personnel | V19 | 0.0512 | |
| | Employment in transportation, storage, and postal industries | V20 | 0.0334 | |

**Table 3. Comprehensive evaluation results of logistics capacity of provinces and cities.**

|  | 2016 | 2017 | 2018 | 2019 | 2020 | Average value |
|---|---|---|---|---|---|---|
| Shanghai | 0.4472 | 0.5216 | 0.5638 | 0.5995 | 0.6275 | 0.5519 |
| Jiangsu | 0.4569 | 0.5097 | 0.5361 | 0.5781 | 0.6099 | 0.5381 |
| Zhejiang | 0.3684 | 0.4304 | 0.4825 | 0.5435 | 0.5769 | 0.4803 |
| Anhui | 0.2144 | 0.2385 | 0.2475 | 0.2561 | 0.2732 | 0.2459 |
| Jiangxi | 0.1022 | 0.0973 | 0.1083 | 0.1406 | 0.1431 | 0.1183 |
| Hubei | 0.1792 | 0.2071 | 0.2286 | 0.2458 | 0.2608 | 0.2243 |
| Hunan | 0.1270 | 0.1428 | 0.1659 | 0.1821 | 0.2053 | 0.1646 |
| Chongqing | 0.1223 | 0.1318 | 0.1422 | 0.1579 | 0.1736 | 0.1456 |
| Sichuan | 0.1724 | 0.1870 | 0.2218 | 0.2387 | 0.2624 | 0.2165 |
| Guizhou | 0.0555 | 0.0564 | 0.0697 | 0.0896 | 0.1058 | 0.0754 |
| Yunnan | 0.0874 | 0.0979 | 0.1019 | 0.1132 | 0.1245 | 0.1049 |

upstream regions such as Sichuan, Guizhou, and Yunnan. Provinces in the middle reaches of the Yangtze River, including Anhui, Hubei, Hunan, and Jiangxi, exhibit fluctuating logistics capability levels, generally at a moderate level.

From a temporal perspective, logistics capabilities across provinces and cities showed an overall upward trend from 2016 to 2020. Shanghai consistently maintained the highest logistics capability level, followed by Zhejiang and Jiangsu. The logistics capabilities of the Yangtze River Economic Belt naturally divide into three core regions: the first, centered around Shanghai, includes Shanghai, Jiangsu, Zhejiang, and Anhui. The second, centered around Hubei, includes Jiangxi, Hubei, and Hunan. The third, centered around Sichuan, includes Chongqing, Sichuan, Yunnan, and Guizhou.

From a spatial perspective, the logistics capabilities of the Yangtze River Economic Belt exhibit an "east high, west low" staggered distribution. The logistics capability level is high in the Yangtze River Delta region, moderate in the central provinces, and low in the western provinces. Shanghai has consistently ranked first in logistics capabilities for five consecutive years, followed by Zhejiang and Jiangsu. As the core region for the development of the Yangtze River Economic Belt, Shanghai, Jiangsu, and Zhejiang lead in infrastructure, economic size, and technological aspects compared to other provinces and cities. Significant achievements in logistics capability development have been made in Anhui, Jiangxi, Hubei, Hunan, Chongqing, and Sichuan. The logistics capabilities of provinces such as Yunnan and Guizhou are relatively lower, but there is a continuous improvement in their levels. In terms of the average evaluation results of logistics capabilities for each province and city, Shanghai's capability is five times that of provinces like Yunnan and Guizhou. This indicates significant imbalances and gaps in the development of logistics capabilities across regions in the Yangtze River Economic Belt.

## The result of global spatial autocorrelation analysis

On the basis of evaluating the logistics capacity of the provinces in the Yangtze River Economic Belt, due to the spatial correlation of the logistics capacity level between different regions, this paper adopts the panel data of the provinces (municipalities) in the Yangtze River Economic Belt from 2016 to 2020, and applies the stata16.0 software to obtain the global Moran's I index of the corresponding years to test the spatial dependence of the data variable.formula for the global Moran's I index (9).

Table 4 shows the results of the global Moran's I index of the logistics capacity of the Yangtze River Economic Belt from 2016 to 2020 under the conditions of neighbor space weight matrix, inverse distance space weight matrix and economic distance space weight matrix. It

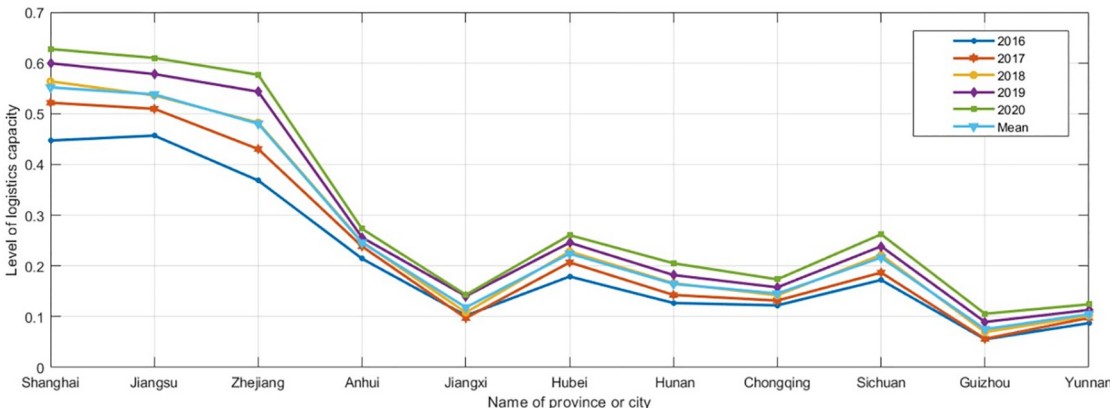

**Fig 1. Line graph of logistics capacity level results.**

can be seen that the global Moran's I indexes are all significant at the 1% level, indicating that there is a significant positive spatial correlation between the logistics capacity of the provinces in the Yangtze River Economic Belt. The consistent conclusions obtained under the three spatial weight matrix conditions further indicate that the choice of spatial econometric model in this paper to analyze the regional physical capacity of the Yangtze River Economic Belt is reasonable.

## Local spatial autocorrelation analysis

Global Moran's Index can illustrate the overall spatial correlation of logistics capacity across provinces within the Yangtze River Economic Belt. To depict the local spatial disparities in the logistics capacity level more clearly, Stata 16.0 software was utilized. The construction of an adjacency spatial weight matrix based on the Queen principle was employed, and local Moran scatterplots were generated. Fig 2 display the analysis of local spatial autocorrelation of logistics capacity within the Yangtze River Economic Belt for the years 2016, 2018, and 2020. It can be observed that the majority of provinces' scatterplots are situated in the first and third quadrants, indicating positive spatial correlation. Provinces in the first quadrant primarily belong to the more economically developed regions in the eastern part of the Yangtze River Economic Belt, such as Shanghai, Zhejiang, and Jiangsu. Provinces in the third quadrant mainly represent the relatively less developed regions in the western part of the Yangtze River Economic Belt, including Yunnan, Guizhou, and Chongqing.

The formation of such distribution pattern is closely related to the socio-economic and geographical factors of each province. Yangtze River Economic Belt regional economic development formed the Yangtze River Delta city cluster, the middle reaches of the Yangtze River city

**Table 4. Global Moran's I of logistics capacity in the Yangtze River Economic Belt, 2016–2020.**

| Year | Logistics Capacity | | |
|---|---|---|---|
| | $W_{0-1}$ | $W_{inv}$ | $W_{eco}$ |
| 2016 | 0.672*** | 0.247*** | 0.742*** |
| 2017 | 0.665*** | 0.248*** | 0.746*** |
| 2018 | 0.642*** | 0.243*** | 0.771*** |
| 2019 | 0.651*** | 0.246*** | 0.815*** |
| 2020 | 0.636*** | 0.240*** | 0.818*** |

Note: ***, **, * indicate significant at the 1%, 5%, and 10% levels, respectively.

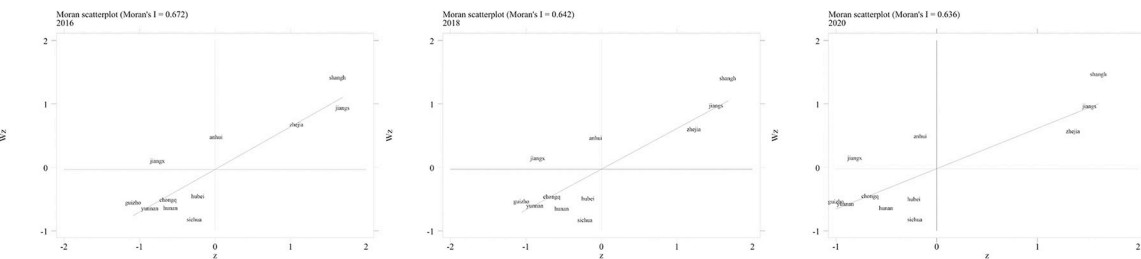

**Fig 2. Local Moran Index scatter plot of logistics capacity in 2016, 2018 and 2020.**

cluster, Chengdu-Chongqing city cluster. Located in the lower reaches of the Yangtze River, the Yangtze River Delta city cluster mainly includes Shanghai, Jiangsu, Zhejiang and Anhui provinces, with strong economic strength. The urban agglomerations in the middle reaches of the Yangtze River mainly include Hubei, Hunan and Jiangxi, which are geographically close and have frequent economic and trade exchanges. The Chengdu-Chongqing city cluster mainly includes Chengdu, Chongqing, Guizhou and other provinces with similar economic development levels.

In addition to the economic factors of each province and city, the geographical environment factors are also important factors. The Yangtze River originates from the Tanggula Mountains, with a total length of 6,300 kilometers, and enters the sea east of Chongming Island in Shanghai. The upper reaches are dominated by high mountains and valleys, the middle reaches are dominated by mountains and hills, and the lower reaches are dominated by plains. From the perspective of land transportation, high mountains and valleys, mountain and hilly terrain, it is difficult to build Bridges and roads, which is not conducive to logistics transportation and regional economic and trade exchanges. From the perspective of waterway transportation, the upstream river drop is large, the river channel is narrow, the water flow is rapid, which is not conducive to waterway transportation, and the downstream area is flat, the water surface is broad, and the water flow is gentle, which is conducive to waterway transportation. Lead to the current spatial distribution pattern.

## Spatial measurement results analysis

### Control variable selection

In conjunction with the regional logistics capacity index system, this study selects road network density (V5) as a control variable from the perspective of logistics infrastructure design. From the perspective of logistics scale, transportation, storage, and postal services' gross value added (V9), port throughput (V10), and waterway freight turnover (V13) are chosen as control variables. Technological and human resources are significant factors influencing logistics capacity. Therefore, this study selects technological market turnover (V16) and the number of employees in transportation, storage, and postal services (V20) as control variables. Descriptive statistics of the dependent and control variables are presented in Table 5.

To avoid multi-collinearity issues resulting from strong correlations among macroeconomic variables, this research conducts variance inflation factor (VIF) tests on the control variables. According to the VIF test results, the VIF value in this study's model is 4.65, significantly lower than 10, indicating the absence of multi-collinearity.

### Testing and selection of spatial econometric models

Before assessing spatial spillover effects, LM-lag, LM-err, R-LM-lag, and R-LM-err methods are employed to test the spatial lag model and spatial error model. The results, as shown in

**Table 5. Descriptive statistics of dependent and control variables.**

| Variable | Obs | Mean | Std. Dev. | Min | Max |
|---|---|---|---|---|---|
| y | 55 | 0.261 | 0.175 | 0.056 | 0.628 |
| V5 | 55 | 1.439 | 0.521 | 0.624 | 2.494 |
| V9 | 55 | 1470.62 | 670.336 | 328.41 | 3239.92 |
| V10 | 55 | 5.860 | 8.133 | 0.002 | 29.655 |
| V13 | 55 | 4885.223 | 7677.058 | 7.18 | 32094.55 |
| V16 | 55 | 575.455 | 533.972 | 20.444 | 2087.847 |
| V20 | 55 | 28.976 | 11.850 | 119 | 51.1 |

Table 6, indicate significance at the 1% or 5% level under the condition of the adjacency spatial weight matrix, suggesting consideration for the Spatial Durbin model. The Hausman test results indicate that under three spatial weight matrix conditions, the appropriate choice is the Spatial Durbin model with fixed effects. Under the geographic distance weight matrix, LM-err-test did not pass the significance test among the four tests, indicating suitability for the spatial lag model analysis. Under the economic distance weight matrix, R-LM-err-test did not pass the significance test among the four tests, indicating suitability for the spatial lag model analysis. Wald-lag test, LR-lag test, Wald-err test, and LR-err test all show significance at the 1% level, indicating the selection of a spatial model that encompasses both endogenous and exogenous interaction effects.

## Logistic capability spatial spillover effects

From a spatial perspective, the spatial autoregressive coefficient ($\rho$) is significantly positive at the 1% level under three spatial weight matrices. This indicates a significant endogenous interaction effect in the regional logistic capability levels. The well-connected road network system, closely linked water transport system, and free-flowing human resources and technology among provinces in the Yangtze River Economic Belt contribute to the enhancement of regional logistic capability. This facilitates the optimization of regional logistic resources and their efficient utilization. From standardized data, it can be observed that a 1% increase in logistic capability in adjacent provinces leads to a 0.54%, 0.557%, and 0.776% increase in local city logistics levels, as shown in Table 7.

**Table 6. Spatial econometric model tests.**

| | W$_{0-1}$ | | W$_{inv}$ | | W$_{eco}$ | |
|---|---|---|---|---|---|---|
| | | *p*-value | | *p*-value | | *p*-value |
| LM-lag test | 39.494 | 0.000 | 13.092 | 0.000 | 31.989 | 0.000 |
| R-LM-lag test | 31.530 | 0.000 | 22.198 | 0.000 | 8.934 | 0.003 |
| LM-err-test | 12.189 | 0.000 | 1.811 | 0.178 | 23.292 | 0.000 |
| R-LM-err-test | 4.225 | 0.040 | 10.916 | 0.001 | 0.238 | 0.626 |
| Moran's I | 3.843 | 0.000 | 1.734 | 0.083 | 4.946 | 0.000 |
| Wald-lag test | 31.91 | 0.000 | 29.37 | 0.000 | 32.74 | 0.000 |
| LR-lag test | 26.63 | 0.003 | 25.66 | 0.004 | 26.77 | 0.000 |
| Wald-err test | 39.93 | 0.000 | 36.74 | 0.000 | 31.07 | 0.000 |
| LR-err test | 28.88 | 0.000 | 28.18 | 0.000 | 26.27 | 0.000 |
| Hausman test | -48.55 | | -3.04 | | 328.15 | 0.000 |

**Table 7. Spatial Durbin model estimation results.**

|  | W$_{0-1}$ |  | W$_{inv}$ |  | W$_{eco}$ |  |
|---|---|---|---|---|---|---|
|  |  | t-statistic |  | t-statistic |  | t-statistic |
| lnV5 | 2.490** | (2.16) | 2.084 | (1.56) | 4.935*** | (5.76) |
| lnV9 | -2.699* | (-1.74) | -3.275** | (-2.20) | -0.317 | (-0.41) |
| lnV10 | 0.605* | (1.80) | 0.391 | (1.42) | 0.295 | (1.48) |
| lnV13 | 3.775 | (1.58) | 4.612 | (1.51) | 1.900 | (0.95) |
| lnV16 | 1.137* | (1.88) | 0.804 | (0.80) | 1.854*** | (3.22) |
| lnV20 | 6.245 | (1.02) | 8.894 | (1.50) | 4.480 | (1.35) |
| W.lnV5 | -0.143 | (-0.03) | 0.678 | (0.06) | -4.372*** | (-4.67) |
| W.lnV9 | 6.900** | (2.10) | 0.581 | (0.18) | -1.479* | (-1.89) |
| W.lnV10 | -1.587** | (-1.98) | -1.026 | (-0.42) | -0.237 | (-0.88) |
| W.lnV13 | 3.767 | (0.97) | 6.724 | (0.85) | 1.429 | (0.55) |
| W.lnV16 | 1.235 | (1.20) | 3.349* | (1.94) | -0.00722 | (-0.02) |
| W.lnV20 | 11.68 | (1.01) | 13.82* | (1.70) | -8.220*** | (-4.02) |
| ρ | 0.540*** | (4.52) | 0.557*** | (5.94) | 0.776*** | (24.94) |
| $R^2$ | 0.730 |  | 0.750 |  | 0.686 |  |
| log-likelihhod | -100.7 |  | -104.4 |  | -87.76 |  |

## Spatial spillover effects analysis

In the presence of spatial spillover effects, a change in a factor not only affects the variation in local logistic capability but also influences the logistic capability in neighboring areas. Following LeSage's research, the total effect of explanatory variables is divided into direct and indirect effects [36]. The direct effect represents the impact of local explanatory variables on the local dependent variable, while the indirect effect signifies the influence of local explanatory variables on the dependent variables of neighboring areas.

As per Table 8, under three spatial weights, road network density (V5) positively influences the improvement of local logistic capability levels. This suggests that higher regional road network density leads to a more robust logistics infrastructure and, consequently, higher logistic capability levels. Simultaneously, local road network construction serves as a positive "demonstration effect" for economically similar cities within the Yangtze River Economic Belt, encouraging surrounding provinces to increase investment in road network construction, enhance the network system, and fully leverage logistic capabilities to drive economic development.

**Table 8. Estimation of direct and indirect effects.**

|  | W$_{0-1}$ |  |  | W$_{inv}$ |  |  | W$_{eco}$ |  |  |
|---|---|---|---|---|---|---|---|---|---|
|  | Direct effect | Indirect effect | Total effect | Direct effect | Indirect effect | Total effect | Direct effect | Indirect effect | Total effect |
| lnV5 | 4.840** | 6.290 | 11.13 | 4.139 | 11.83 | 15.97 | 4.529* | -1.549 | 2.980 |
|  | (2.457) | (6.539) | (7.927) | (3.158) | (19.78) | (22.02) | (2.440) | (4.060) | (6.242) |
| lnV9 | -4.306*** | -2.439 | -6.745 | -5.786*** | -14.94 | -20.73 | -6.535*** | -13.05*** | -19.58*** |
|  | (1.166) | (3.713) | (4.397) | (2.175) | (11.91) | (13.87) | (1.407) | (3.072) | (4.303) |
| lnV10 | 0.348 | -1.154* | -0.807 | 0.114 | -0.629 | -0.515 | 0.464* | 0.350 | 0.814 |
|  | (0.269) | (0.660) | (0.745) | (0.415) | (2.344) | (2.626) | (0.266) | (0.639) | (0.828) |
| lnV13 | 8.157*** | 20.78*** | 28.94*** | 7.728*** | 42.61** | 50.33** | 5.159*** | 7.576* | 12.74** |
|  | (1.513) | (4.638) | (5.133) | (2.465) | (18.02) | (19.80) | (1.531) | (4.229) | (5.301) |
| lnV16 | -1.271* | -5.238*** | -6.509*** | -1.515* | -7.869** | -9.385** | 0.136 | -1.231 | -1.094 |
|  | (0.700) | (1.808) | (2.252) | (0.902) | (3.969) | (4.585) | (0.645) | (1.117) | (1.681) |
| lnV20 | 15.39*** | 32.29** | 47.68*** | 20.82*** | 95.89** | 116.7*** | 13.63*** | 12.90* | 26.52** |
|  | (3.986) | (13.38) | (15.54) | (6.141) | (40.42) | (45.28) | (4.047) | (7.757) | (11.19) |

The Gross Value of Production (V9) in the transportation, warehousing, and postal industry shows a significantly negative impact on both local and surrounding area logistic capabilities under three spatial weight matrices. Typically, higher production values in the transportation, warehousing, and postal industry indicate more developed regional logistics. However, the study results reveal that higher production values significantly limit the improvement of logistic capability in the Yangtze River Economic Belt. This can be explained by considering the positive impact of the number of employees in the transportation, warehousing, and postal industry (V20) on regional logistic capability. An increase in the workforce promotes the enhancement of regional logistic capability, but the accompanying rise in labor costs counteracts the increase in production values, negatively affecting the improvement of regional logistic capability.

Port throughput (V10) exhibits a positive but non-significant effect on regional logistic capability levels under three spatial weight matrices. The research results indicate that an increase in port throughput in the local area positively influences logistic capability. However, under adjacent spatial weight matrices and geographical distance weight matrices, the indirect effect is negative, suggesting that an increase in local port throughput negatively impacts the logistic capability levels of other regions. This is consistent with reality; within a certain time frame, an increase in port throughput in one region inevitably reduces throughput in other regions.

Water freight turnover (V13) demonstrates a positive promotion effect on regional logistic levels under three spatial weight matrices. Water freight turnover equals freight volume multiplied by freight distance. As the Yangtze River Economic Belt's regional economy grows and opens up, the regional water freight turnover continues to increase, contributing to the rise in regional logistic capability. Additionally, the increase in local water freight turnover has positive "demonstration" and "competition" effects on surrounding areas, prompting neighboring provinces to invest in water transport, thereby boosting logistic capability.

An increase in the turnover of the technology market (V16) shows a significantly negative impact on the improvement of logistic capability in both the local area and neighboring provinces under three spatial weight matrices. The development of the technology market reflects the prosperity of the region in developing technology goods to their widespread application. However, the high cost of technology development, transfer, and services, coupled with the inflexible research capabilities of researchers, negatively affects the improvement of logistic capability levels. Moreover, only a small portion of scientific technology in the technology market serves the logistics industry, limiting its contribution to the enhancement of logistic capability.

An increase in the number of employees in the transportation, warehousing, and postal industry (V20) shows a significantly positive impact on the improvement of both local and neighboring area logistic capabilities under three spatial weight matrices. This industry falls under the production-oriented service sector, and its production process demands a substantial workforce. Additionally, the current level of smart technology in the transportation, warehousing, and postal industry in the provinces and cities along the Yangtze River Economic Belt is not high, necessitating increased labor input for industry development.

## Robustness test

To ensure the robustness of the research structure, this study first estimated two models, the Spatial Autoregressive Model (SAR) and the Spatial Error Model (SEM), based on three spatial weight matrices (Table 9). Subsequently, the spatial Durbin model was estimated with a lag of one period for the dependent variable (columns 4, 7, and 10 in Table 9), and the corresponding

**Table 9.  Regression results of spatial lag, spatial error, and spatial Durbin models with a one-period lag for the dependent variable.**

|  | $W_{0-1}$ | | | $W_{inv}$ | | | $W_{eco}$ | | |
|---|---|---|---|---|---|---|---|---|---|
|  | SLM | SEM | SDM | SLM | SEM | SDM | SLM | SEM | SDM |
| lnV5 | -2.762** | -4.191** | -5.057*** | -3.425*** | -4.458** | -5.225*** | -1.186 | -0.0401 | -0.0715 |
| lnV9 | 0.382 | -0.842 | -0.450 | -0.216 | -1.496 | -1.058 | 1.946** | 1.741*** | 0.722 |
| lnV10 | 0.390 | 0.528* | 0.561** | 0.494 | 0.588 | 0.252 | 0.182 | 0.364 | 0.345 |
| lnV13 | 1.110 | 2.618 | 3.450 | 2.953 | 4.209 | 4.296 | 1.070 | 1.656 | 1.595 |
| lnV16 | 2.664** | 0.865 | 1.345 | 1.805 | -0.128 | 0.747 | 2.457*** | 1.564** | 2.148** |
| lnV20 | 4.611 | 6.176 | 10.27 | 6.522 | 8.603 | 16.05* | 4.195 | 8.806** | 6.072 |
| W.lnV5 |  |  | 5.070 |  |  | -1.965 |  |  | -0.161 |
| W.lnV9 |  |  | 6.191 |  |  | 5.323** |  |  | -2.612*** |
| W.lnV10 |  |  | -0.883 |  |  | -1.496 |  |  | -0.379 |
| W.lnV13 |  |  | -2.462 |  |  | 1.681 |  |  | -0.228 |
| W.lnV16 |  |  | 2.992** |  |  | 5.925** |  |  | -0.132 |
| W.lnV20 |  |  | 15.98 |  |  | 39.63 |  |  | -8.764*** |
| $\rho$ | 0.701*** |  | 0.517*** | 0.827*** |  | 0.600*** | 0.709*** |  | 0.771*** |
| $\lambda$ |  | 0.850*** |  |  | 0.884*** |  |  | 0.857*** |  |
| $R^2$ | 0.836 | 0.738 | 0.844 | 0.793 | 0.707 | 0.771 | 0.852 | 0.764 | 0.733 |
| log-likelihhod | -80.829 | -82.520 | -75.059 | -80.989 | -81.765 | -74.717 | -73.719 | -76.073 | -69.159 |

estimates of direct and indirect effects were obtained (Table 10). The results of these two tests demonstrate spatial spillover effects of logistics capability, and there were no significant changes in the coefficient signs and significance of each explanatory variable. This indicates that the conclusions drawn in the previous sections are robust.

## Policy implications

The development of regional economy is an important part of building a dynamic and sustainable national economic system, and has a far-reaching impact on the realization of comprehensive, coordinated and sustainable development. The development of regional economy plays a positive role in optimizing regional resource allocation, promoting technological innovation, narrowing regional development gap and improving residents' living standards. Based on the research conclusions, the following suggestions are put forward to promote the improvement of regional logistics capacity and regional economic development of the Yangtze River Economic Belt.

First, strengthen regional coordination and interaction to promote the transformation and upgrading of industrial structure. The Yangtze River Economic Belt is a "chess game" of regional development, adhering to the five development concepts of "innovative, coordinated,

**Table 10.  Estimates of direct and indirect effects in the spatial Durbin model with a one-period lag for the dependent variable.**

|  | $W_{0-1}$ | | | $W_{inv}$ | | | $W_{eco}$ | | |
|---|---|---|---|---|---|---|---|---|---|
|  | Direct effect | Indirect effect | Total effect | Direct effect | Indirect effect | Total effect | Direct effect | Indirect effect | Total effect |
| lnV5 | -3.001 | 2.376 | -0.626 | -2.613 | 7.118 | 4.505 | -1.614 | -1.942 | -3.557 |
| lnV9 | -2.172 | -1.766 | -3.937 | -4.745** | -15.15 | -19.90 | -3.855*** | -9.067*** | -12.92*** |
| lnV10 | 0.689** | 0.315 | 1.004 | 0.420 | 1.453 | 1.873 | 0.735*** | 1.262** | 1.997*** |
| lnV13 | 7.174*** | 12.43** | 19.61*** | 6.004** | 28.12** | 34.13** | 4.508*** | 5.491 | 9.999** |
| lnV16 | -1.121 | -4.699* | -5.820* | -1.551 | -6.285* | -7.836** | -0.866 | -2.424** | -3.291* |
| lnV20 | 15.98*** | 34.15** | 50.13*** | 19.49*** | 71.78** | 91.27*** | 15.46*** | 16.39*** | 31.86*** |

green, open and shared", giving full play to the guiding role of the government in the process of regional economic integration, and promoting coordinated and interconnected regional economic development.Shanghai adheres to the five development concepts, and jointly approved and issued the "Spatial Collaborative Planning of Shanghai Metropolitan Area" with Jiangsu and Zhejiang provinces. We will guide the development direction and construction priorities of large, small and medium-sized cities in different categories, and form a spatial pattern of urbanization with balanced, division of labor and cooperation and complete functions, reflecting the new development concept of "extensive consultation, joint construction and shared benefits". In addition, the Yangtze River Delta region, especially the core region with Shanghai as the center, is the region with the most active economic development, the highest degree of openness and the strongest innovation ability, playing a core role and assuming the role of explorers and pioneers.

Second, strengthen the division of labor and cooperation in the industrial chain of the whole basin, and give full play to the advantages of each region. Give full play to the positive spillover and radiation driving role of high-level logistics areas, the Yangtze River Delta region to give full play to the "leader" role, strengthen technology and management innovation, relying on the international port logistics hub to connect the international and domestic. The middle and upstream regions actively undertake industrial transfer, and the upstream regions give full play to their rich resources and environmental advantages to explore new ways of development. The middle and lower reaches of the Yangtze River economic belt are interconnected, and the division of labor and cooperation of the industrial chain of the whole basin are realized for joint and balanced development. Provinces in the Yangtze River Economic Belt have actively integrated into the Belt and Road Initiative and actively carried out foreign trade. Chengdu has built an international three-dimensional network extending in all directions. The opening quantity and operation quality of international air routes provide important support for Chengdu's opening up and internationalization construction. Close exchanges with international sister cities make the construction of the Belt and Road gradually mature, constantly expand Chengdu's international influence, and help Chengdu to step in the forefront of inland opening up.

Third, strengthen the construction of logistics infrastructure and smooth logistics channels. The provinces and cities in the Yangtze River Economic Belt should take the Yangtze River as a link to smooth the shipping system, improve port facilities and channel depth, so as to adapt to the passage of large cargo ships, improve the efficiency of water transportation, and give full play to the role of the "golden waterway". Open up the land transport network, smooth transport system, investment in the development of efficient transport networks, including roads, railways, water transport, etc., to ensure smooth logistics. Develop modern warehousing and logistics parks, build multimodal transport hubs, and promote the collaborative operation of different modes of transport to improve transport efficiency. Develop new logistics technologies, support regional school-enterprise cooperation, integration of production and education, promote the transformation of scientific and technological achievements, smooth the flow of business information channels, and promote the improvement of global logistics capabilities through informatization and digitalization.

Fourth, the provinces in the eastern part of the Yangtze River Economic Belt have relatively high logistics capacity. In promoting the improvement of regional logistics capacity, more attention should be paid to optimizing port and shipping networks, developing modern logistics parks, introducing advanced technologies, improving logistics efficiency and service level, while strengthening international logistics cooperation, improving the docking level with the international market, and promoting the development of cross-border e-commerce and international logistics.

Fifth, the logistics capacity of the central and western provinces of the Yangtze River Economic Belt is relatively weak. In promoting the improvement of regional logistics capacity, we should strengthen the construction of inland ports, improve the functions of inland ports on the Yangtze River, increase the transportation capacity of containers and bulk cargo, and improve the status of logistics hubs in inland cities. Develop railway logistics, optimize railway transport channels, improve transport efficiency, and strengthen railway connectivity with other provinces. Promote industrial collaboration, promote the deep integration of logistics and industry, support the coordinated development of upstream and downstream of the industrial chain, and improve the efficiency of the entire industrial chain. Cities such as Chengdu and Chongqing have expanded their logistics networks, strengthened logistics cooperation between the western region and Central and West Asia, expanded international logistics channels, and promoted cross-border trade. The provinces of Hubei, Hunan and Anhui have actively formulated and introduced policies to support the development of the logistics industry, providing tax and financial support to reduce the operating costs of logistics enterprises.

## Conclusions

The regional economic development holds significant importance for a country or region, influencing overall societal prosperity and the well-being of its people. The logistics industry plays a crucial role in facilitating regional economic development. Serving as a vital link connecting producers, consumers, and various components of the supply chain, it has a direct and profound impact on the prosperity and advancement of regional economies. There are many factors that affect regional logistics ability, such as logistics infrastructure, regional economic development level, logistics technology and talent factors. The purpose of this study is to establish a comprehensive model with comprehensive indicators to show the changes of regional logistics capacity in the Yangtze River Economic Belt from two dimensions of time and space. The results show that the level of logistics capacity in each province is rising continuously in terms of time, and presents a distribution of "high in the east and low in the west" in terms of space. The provinces with higher regional logistics capacity are mainly in the economically developed areas in the east, such as Shanghai, Jiangsu and Zhejiang. The logistics capacity of the western provinces is relatively low, such as Yunnan and Guizhou, while the logistics capacity of the central provinces is between the eastern and western provinces. In order to further explore the factors affecting the spatial pattern of regional logistics capacity, the adjacency space weight matrix, geographical distance weight matrix and economic distance weight matrix were selected to build a spatial econometric model to analyze the influencing factors and spatial spillover effects of regional logistics capacity. The analysis results show that the factors such as road network density, port throughput, water freight turnover, transportation, storage and postal industry employment will not only affect the logistics capacity of the region, but also have a spillover effect on the material capacity of neighboring provinces in the Yangtze River Economic Belt. Based on this, the policy suggestions are provided for the provinces to effectively improve the regional logistics capacity.

It should be noted that this study also has some limitations. First, it is limited by the availability of data. This paper only uses the panel data of provinces and cities in the Yangtze River Economic Belt from 2016 to 2020 for empirical research. Future research may consider using the data of prefecture-level cities to get more specific conclusions and put forward more specific policy recommendations. Second, there is no unified standard for the evaluation of logistics capability. Although 20 indicators were selected to measure the level of logistics capacity in various provinces and cities, it is difficult to fully consider the influencing factors of logistics

capacity in the research due to the extensive content of logistics capacity and the difficulty in obtaining some measurement indicators. Therefore, how to accurately measure the regional logistics capability is still a problem that needs to be solved in the future. Finally, in the analysis of the spatial spillover effect of the regional logistics capacity of the Yangtze River Economic Belt, only some factors are selected for spatial factor analysis, which may have the influence of spatial autocorrelation of unobserved factors, resulting in missing variables bias. Relevant influencing factors can be further enriched in the subsequent research.

## Author Contributions

**Methodology:** He Yinnan.

**Software:** He Yinnan, Wang Biao.

**Writing – original draft:** Li Fanghu.

**Writing – review & editing:** Li Fanghu.

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
