## [Decision Letter · Decision Letter 0]

17 Oct 2023

PONE-D-23-24153Logistics capacity measurement and spatial-temporal characteristics analysis: A case study of the Yangtze River Economic Belt in ChinaPLOS ONE

Dear Dr. Yinnan,

Thank you for submitting your manuscript to PLOS ONE. After careful consideration, we feel that it has merit but does not fully meet PLOS ONE’s publication criteria as it currently stands. Therefore, we invite you to submit a revised version of the manuscript that addresses the points raised during the review process.

We look forward to receiving your revised manuscript.

Kind regards,

Qunxi Gong

Academic Editor

PLOS ONE

“Independent Research fund of Joint National-Local Engineering Research Centre for Safe and Precise Coal Mining（Anhui University of Science and Technology）（NO. EC2021013）

Anhui Province quality engineering teaching research project（NO. 2022jyxm1437“

“yes”

6. We note that Figures 2 and 3 in your submission contain [map/satellite] images which may be copyrighted. All PLOS content is published under the Creative Commons Attribution License (CC BY 4.0), which means that the manuscript, images, and Supporting Information files will be freely available online, and any third party is permitted to access, download, copy, distribute, and use these materials in any way, even commercially, with proper attribution. For these reasons, we cannot publish previously copyrighted maps or satellite images created using proprietary data, such as Google software (Google Maps, Street View, and Earth). For more information, see our copyright guidelines: http://journals.plos.org/plosone/s/licenses-and-copyright.

a. You may seek permission from the original copyright holder of Figures 2 and 3 to publish the content specifically under the CC BY 4.0 license. 

Reviewers' comments:

Reviewer's Responses to Questions

**Comments to the Author**

1. Is the manuscript technically sound, and do the data support the conclusions?

Reviewer #1: Partly

Reviewer #2: Partly

2. Has the statistical analysis been performed appropriately and rigorously? 

Reviewer #1: Yes

Reviewer #2: Yes

3. Have the authors made all data underlying the findings in their manuscript fully available?

Reviewer #1: Yes

Reviewer #2: No

4. Is the manuscript presented in an intelligible fashion and written in standard English?

Reviewer #1: No

Reviewer #2: Yes

5. Review Comments to the Author

Reviewer #1: The authors measure the logistics capacity of Yangtze River Economic Belt from 2016 to 2020 using entropy weighted TOPSIS approach, and analyze the spatio-temporal characteristics through spatial autocorrelation models. The manuscript provides a detailed analysis process and extensive results. However, the research significance and innovations are not fully demonstrated. Major revisions are needed to enhance the suitability for publication. The main comments are as follows:

1 Introduction

(1) The author does not clearly explain the innovation of this paper. The innovativeness of this study needs to be highlighted(Page 5). Elaborating the new methods or perspectives adopted in this study would help justify the value and advancement of this work. For example, adopting an comprehensive indicator system to measure provincial logistics capacity.

(2)The literature review mainly reviews the relevant research on logistics capability from the micro and macro aspects (Page 5-6). It can supplement the summary of existing regional logistics capacity evaluation studies, such as the evaluation dimensions, index system and research area of different studies, and clarify the differences between this study and existing studies.

2 Methods

(1) The correspondence between the index system and the dimensions of logistics capability assessment was not established. (Page 10). The authors should explain the rationale behind choosing these specific indicators and how they connect with logistics capacity.

(2) The phenomenon of high correlation between indicators but lack of processing may lead to information redundancy(Page 10). For example, Per capita freight volume and per capita port throughput are correlated to some extent, especially for regions where ports are the main logistics hubs.

3 Results

(1)For the spatial autocorrelation analysis results (Page 17), consider relating the identified spatial patterns to geographic and socioeconomic factors of the provinces, to help explain the clustering.

(2) The multiple linear regression model (Page 19-20) may be overly simplistic. For example, it does not consider spatial effects (spatial autocorrelation/spatial dependence) and individual heterogeneity across provinces. Advanced econometric modeling techniques could help address these issues and enhance the explanatory power. This includes spatial autoregressive models to account for spatial interaction effects, spatial error models for spatial dependence in residuals, as well as panel models with fixed/random effects to control for unobserved provincial characteristics.

4 Discussion

(1) In summarizing the research findings (Page 20), the core contributions and implications of this study could be further accentuated, instead of just restating the results.

(2) The policy recommendations (Page 20) could be more specific, like tailored strategies for different provinces to have practical utility.

5 Others

(1)Most of the references are very old and lack the tracking of recent research frontiers. It is suggested to supplement the relevant literature in the last five years, especially the new progress of logistics development in the Yangtze River region, so as to highlight the innovation of the research.

Reviewer #2: The author uses TOPSS method of entropy weight to measure the logistics capacity of the Yangtze River Economic Belt, analyzes the spatial and temporal distribution characteristics of the logistics capacity of the region by drawing the spatial and temporal evolution of the logistics capacity of the region, and gives suggestions for future development. The research topic is interesting and of great significance for the target area. However, the current manuscript has several problems and does not allow for professional publication. I sincerely suggest that the author carefully revise and prepare for future submissions. My suggestions and comments on the manuscript are as follows:

The main problem is that the current manuscript is imprecise, unclear, and contains a large number of punctuation errors. An example of this situation is the sentence on page 11 lines 27 through 29:

“Although there have been many studies on regional logistics capability, the current research on logistics capability mainly focuses on the logistics capability of enterprises and ports, with few studies focusing on regional logistics capability“

I think this sentence is contradictory in its presentation

On the other hand, the basic requirement of a scientific article is to use correct and precise mathematical expressions. However, this text still exists

a) The equation numbering format is not uniform, for example, the numbering format of Formula ([Disp-formula pone.0303200.e012]) on page 15 is not uniform with other equation numbering formats on this page.

b) Lack of definition, such as x+ and x- variables on page 14.

These incorrect contents greatly reduce the readability of this article. The existing manuscripts are considered rich in research results, but need to have the right content for professional publication. Therefore, I sincerely advise the author to take enough time to revise carefully to prevent inaccurate expression and content.

Here are some other tips:

1) the sentence "In the 1980s, in lines 13 to 14 and 22 to 23 on page 9, scholars carried out extensive and in-depth research on the logistics capability of enterprises [3-5]. “, “Enterprises can use and can allocate resources reasonably to achieve maximum efficiency and improving customer care needs [6,7,8,9,11]. "Literature reviews should avoid cluster citations, recommend a brief description of the novel contributions of each paper cited, and reduce the number of references that are sub-related to the research direction of the article."

2) There are errors in the format of references, refer to the journal's reference style for the exact format of these references, as well as the use of punctuation and capitalization.

3) In the conclusion, in addition to summarizing the actions and results taken, please strengthen the explanation of their significance. It is recommended to use quantitative reasoning for comparison with appropriate benchmarks, especially those derived from previous work.

6. PLOS authors have the option to publish the peer review history of their article (what does this mean?). If published, this will include your full peer review and any attached files.

Reviewer #1: No

Reviewer #2: No

---

## [Author Response · Author response to Decision Letter 0]

19 Dec 2023

Dear Reviewer,

Thank you for providing us with these suggestions to revise and improve the quality of our article. These comments were all valuable and very helpful.

We regret that our paper does not meet your expectations. We have reread the article and made comprehensive changes. We still implore you to give us an opportunity, which is crucial for us.

We have carefully read your comments and have tried our best to revise our manuscript based on these valuable and useful comments. Please find attached the revised version and the supplemental files, which we would like to submit for your kind consideration.

---

## [Decision Letter · Decision Letter 1]

1 Mar 2024

PONE-D-23-24153R1Analysis of logistics capacity, influencing factors and spatial spillover effect in Yangtze River Economic BeltPLOS ONE

Dear Dr. Yinnan,

Thank you for submitting your manuscript to PLOS ONE. After careful consideration, we feel that it has merit but does not fully meet PLOS ONE’s publication criteria as it currently stands. Therefore, we invite you to submit a revised version of the manuscript that addresses the points raised during the review process.

We look forward to receiving your revised manuscript.

Kind regards,

Qunxi Gong

Academic Editor

PLOS ONE

Journal Requirements:

Reviewers' comments:

Reviewer's Responses to Questions

**Comments to the Author**

1. If the authors have adequately addressed your comments raised in a previous round of review and you feel that this manuscript is now acceptable for publication, you may indicate that here to bypass the “Comments to the Author” section, enter your conflict of interest statement in the “Confidential to Editor” section, and submit your "Accept" recommendation.

Reviewer #1: All comments have been addressed

2. Is the manuscript technically sound, and do the data support the conclusions?

Reviewer #1: Yes

3. Has the statistical analysis been performed appropriately and rigorously? 

Reviewer #1: Yes

4. Have the authors made all data underlying the findings in their manuscript fully available?

Reviewer #1: Yes

5. Is the manuscript presented in an intelligible fashion and written in standard English?

Reviewer #1: Yes

6. Review Comments to the Author

Reviewer #1: The authors have made a commendable effort to address the concerns raised in the initial review. They have clarified the innovation of the paper by highlighting the comprehensive index system for measuring provincial logistics capacity and the use of spatial econometrics methods to analyze influencing factors and spatial spillover effects. The literature review has been expanded to include a summary of existing regional logistics capacity evaluation studies, which helps to situate this study within the broader field of research.

The authors have also improved the correspondence between the index system and the dimensions of logistics capability assessment by explaining the rationale behind the selection of specific indicators. The issue of high correlation between indicators has been addressed by refining the index system and ensuring that it is more representative of the logistics capacity.

The spatial autocorrelation analysis has been enhanced by relating the identified spatial patterns to geographic and socioeconomic factors of the provinces, which provides a more nuanced understanding of the clustering observed. The multiple linear regression model has been replaced with a spatial econometric model, which accounts for spatial interaction effects and individual heterogeneity across provinces, significantly improving the explanatory power of the analysis.

The discussion section has been strengthened by accentuating the core contributions and implications of the study, rather than merely restating the results. The policy recommendations have been made more specific, tailored to the needs of different provinces, which increases the practical utility of the study.

The authors have also updated the references to include recent research, which helps to highlight the innovation of the study and situates it within the current research frontier.

Overall, the manuscript has been significantly improved, and the authors have shown a willingness to address the concerns raised. However, there are still some areas that could be improved:

1.The authors should ensure that the revised manuscript is free of punctuation errors and hat the references are formatted consistently with the journal's guidelines.

2.The authors should provide a more detailed explanation of the methodology used, especially the rationale behind the selection of the spatial weighting matrix and the interpretation of the spatial econometric model results.

3.The authors should consider discussing the limitations of the study, such as the potential for spatial heterogeneity within provinces and the assumptions made in the spatial econometric model.

Given the improvements and the remaining areas for further work, I recommend the manuscript for publication with minor revisions.

7. PLOS authors have the option to publish the peer review history of their article (what does this mean?). If published, this will include your full peer review and any attached files.

Reviewer #1: No

---

## [Author Response · Author response to Decision Letter 1]

13 Mar 2024

Dear Reviewer,

It is our great honor to receive the comments on my paper from you. Those comments are all valuable and very helpful for revising and improving our paper.

We have studied the comments carefully and have made corrections which we hope meet the approval. Revised portions are marked in blue on the paper. You can find them in the attached files uploading. If there are any errors, please notify us. We will correct them accordingly.

Once again, thank you very much for your comments and suggestions.

Comment 1: The authors should ensure that the revised manuscript is free of punctuation errors and hat the references are formatted consistently with the journal’s guidelines.

Response 1: We thank the reviewer for raising this question. We should do the work that the sentences of articles are smooth, the punctuation is standardized and the format of articles conforms to the requirements of periodical format. We will carefully revise to ensure that such issues are addressed. Thank you very much!

Revise 1:(pages 26-28 lines 853-956)

People's Republic of China. Guiding Opinions of The State Council on Promoting the development of the Yangtze River Economic Belt by relying on the Golden Waterway. 2014,9. Available from: https://www.gov.cn/gongbao/content/2014/content_2758494.htm

People's Republic of China. The National Development and Reform Commission issued a notice on the Layout Plan of the Yangtze River Trunk Crossing Passage (2020-2035). 2020,3. Available from: https://www.gov.cn/zhengce/zhengceku/2020-04/08/content_5500124.htm

Zawawi NFBM, Wahab SA, Mamun AA, et al. Logistics Capability, Information Technology, and Innovation Capability of Logistics Service Providers: Empirical Evidence from East Coast Malaysia. International Review of Management & Marketing, 2017, 7.

Wei HL, Jian HJI. Research on the management process of logistics capabilities in supply Chain. Journal of Chongqing Jiaotong University, 2006.

Bowersox DJ, Closs DJ. Logistical management: the integrated supply chain process(vol.2). 1999.

Daugherty, Patricia J, Pittman, et al. Utilization of time-based strategies: Creating distribution flexibility/responsiveness. International Journal of Operations & Production Management, 1995. doi:10.1108/01443579510080418.

Fawcett SE, Stanley LL, Smith SR. Developing a logistics capability to improve the performance of international operations. Journal of Business Logistics, 1997; 18(2):101-127.

Lynch, Daniel F, Scott B. Keller, John O. The effects of logistics capabilities and strategy on firm performance. Journal of business logistics. 2000; 21(2): 47.

Zhao M, Theodore PS. Interactions between operational and relational capabilities in fast food service delivery. Transportation Research Part E: Logistics and Transportation Review .2003; 39(2): 161-173.

Bharadwaj S, Bharadwaj A, Bendoly E. The Performance Effects of Complementarities Between Information Systems, Marketing, Manufacturing, and Supply Chain Processes. Information Systems Research, 2007; 18(4):437-453. doi:10.1287/isre.1070.0148.

Wu F, Yeniyurt S, Kim D, et al. The impact of information technology on supply chain capabilities and firm performance: A resource-based view. Industrial Marketing Management, 2006; 35(4): 493-504. doi:10.1016/j.indmarman.2005.05.003.

Li JJ. Competitive position, managerial ties, and profitability of foreign firms in China: an interactive perspective. Strategic Direction, 2009; 25(8). doi:10.1108/sd.2009.05625had.008.

Lynch D, Keller S, Ozment J. The effects of logistics capabilities and strategy on firm performance. Journal of business logistics. 2000; 21 (2): 47.

Ackermann J, Egon M. Modelling, planning and designing of logistics structures of regional competence-cell-based networks with structure types. Robotics and Computer Integrated Manufacturing, 2007; 23(6):601-607. doi:10.1016/j.rcim.2007.02.010.

Lu H. A model of integrated regional logistics hub in supply chain. Enterprise Information Systems. 2018; 12(10): 1308-1335.

Guo HP. MLP neural network-based regional logistics demand prediction. Neural Computing and Applications.2021; 33 (9): 3939-3952.

Heng C, Shan YJ, Zeng T. An Analysis of the Development Mode and Influence Effect of Regional Logistics Industry in China. Commercial Research. 2020; 62(5): 55.

18Zhang D, Eglese R, Li S. Optimal location and size of logistics parks in a regional logistics network with economies of scale and CO2 emission taxes. Transport, 2015; 33(1):1-17. doi:10.3846/16484142.2015.1004644.

Zhao W, Zhang J, Zhao L. Research on Evaluation of Regional Logistics Capability under Big Data Background. MATEC Web of Conferences, 2017; 100.

Carlucci F ,Cirà, Andrea, Forte E . Infrastructure and logistics divide: regional comparisons between North Eastern & Southern Italy. Technological & Economic Development of Economy. 2017;1-27.

Huang X, Wang Y, Dai X, et al. Evaluation of port efficiency in shanghai port and busan port based on three-stage DEA model with environmental concerns. Transport . 2019; 35(5):1-8. doi:10.3846/transport.2019.11465.

Hylton PJ, Ross CL. Agglomeration economies' influence on logistics clusters' growth and competitiveness. Regional Studies. 2018;1-12.

Sun T, Nong P, Weng SZ. Comprehensive evaluation of regional logistics development level in Guangxi based on fuzzy matter-element analysis . Logistics Engineering and Management. 2019; 44(06):6-9+13.

Jiang H, Sun TP, Zhuang BN. Determinants of Low-Carbon Logistics Capability Based on Dynamic fsQCA: Evidence from China's Provincial Panel Data. Sustainability. 2023; 15(14):7-12.

Zhang GS, Xiang WM. Research on evaluation of regional logistics capacity of four major urban agglomerations in China. Journal of Chongqing Technology and Business University (Social Science Edition). 1-13 http://kns.cnki.net/kcms/detail/50.1154.c.20231109.1726.006.html.

He XG, Lu GD. The influence of regional logistics capability on the performance of circulation enterprises: Based on empirical evidence of listed companies in circulation industry. Business Economics Research, 2023; 6(21): 26-29.

Shi PZ. Research on evaluation and improvement path of regional logistics capability under new development Concept: A case study of Henan Province. Journal of Henan Polytechnic University (social sciences Edition).2023; 24(04): 31-41.

Pavić, Zlatko, Vedran Novoselac. Notes on TOPSIS method. Int. J. Res. Eng. Sci. 2013;1 (2): 5-12.

Shi T, Si SC, Chan J. The Carbon Emission Reduction Effect of Technological Innovation on the Transportation Industry and Its Spatial Heterogeneity: Evidence from China. Atmosphere. 2021;12(9):23-26.

Aleksy K, Oleksii L, Tetyana P. Spillover Effects of Green Finance on Attaining Sustainable Development: Spatial Durbin Model. Computation. 2023; 11(10): 12-21.

Zhang C. Research on the Economical Influence of the Difference of Regional Logistics Developing Level in China. Journal of Industrial Integration and Management, 2020. doi:10.1142/S2424862220500049.

Guo ZX, Tian Y, Guo XM, et al. Research on Measurement and Application of China's Regional Logistics Development Level under Low Carbon Environment. PROCESSES, 2021,9(12).

Yu N, Xu W, Yu KL. Research on Regional Logistics Demand Forecast Based on Improved Support Vector Machine: A Case Study of Qingdao City under the New Free Trade Zone Strategy. IEEE ACCESS, 2020,8:9551-9564.

Liu J. Development of Regional Logistics Along the One Belt and One Road. Springer Singapore, 2016; 77-101.

China Statistical Yearbook. 2016-2020. Available online: http://www.stats.gov.cn/sj/ndsj/

LeSage J, Pace RK. Introduction to Spatial Econometrics. New York: Chapman and Hall/CRC, 2009.

Comment 2: The authors should provide a more detailed explanation of the methodology used, especially the rationale behind the selection of the spatial weighting matrix and the interpretation of the spatial econometric model results.

Response2: Thank you for your very good advice. We have added a description of the methodology used in the correspondence section. 

Revise 2:(pages 8,9 lines303-320)

The traditional measurement model only studies the influence of individual independent variables on individual dependent variables, while the spatial measurement model further explores the influence of individual independent variables on other individual dependent variables on the basis of traditional measurement. Therefore, the greatest feature of spatial metrology is to fully consider the spatial correlation between cross-sectional units. Spatial correlation refers to the spatial interdependence, mutual restriction, mutual influence and interaction between things and phenomena in different regions. It is the inherent spatial economic attribute of things and phenomena, and the essential attribute of geographical spatial phenomena and spatial processes. The spatial correlation is caused by the proximity of geographical location, the competition and cooperation of individuals at the cross-section level, the group imitation behavior, the spillover effect of economic activities and the measurement error. Therefore, in the actual construction of the model, how to incorporate the spatial effects of economic variables into the model, it is necessary to establish a spatial weight matrix according to some standards. To define the spatial weight, we must first quantify the position of the spatial unit, which usually includes neighborhood space distance, geographical distance, economic distance and industrial distance.

Comment 3: The authors should consider discussing the limitations of the study, such as the potential for spatial heterogeneity within provinces and the assumptions made in the spatial econometric model.

Response 3: Thank you for your suggestion. This is a very good suggestion. According to your suggestion, we have added content to the article about the limitations of the study.

Revise 3:(page25,26 lines 836-851)

It should be noted that this study also has some limitations. First, it is limited by the availability of data. This paper only uses the panel data of provinces and cities in the Yangtze River Economic Belt from 2016 to 2020 for empirical research. Future research may consider using the data of prefecture-level cities to get more specific conclusions and put forward more specific policy recommendations. Second, there is no unified standard for the evaluation of logistics capability. Although 20 indicators were selected to measure the level of logistics capacity in various provinces and cities, it is difficult to fully consider the influencing factors of logistics capacity in the research due to the extensive content of logistics capacity and the difficulty in obtaining some measurement indicators. Therefore, how to accurately measure the regional logistics capability is still a problem that needs to be solved in the future. Finally, in the analysis of the spatial spillover effect of the regional logistics capacity of the Yangtze River Economic Belt, only some factors are selected for spatial factor analysis, which may have the influence of spatial autocorrelation of unobserved factors, resulting in missing variables bias. Relevant influencing factors can be further enriched in the subsequent research.

---

## [Decision Letter · Decision Letter 2]

22 Apr 2024

Analysis of logistics capacity, influencing factors and spatial spillover effect in Yangtze River Economic Belt

PONE-D-23-24153R2

Dear Dr. Yinnan,

We’re pleased to inform you that your manuscript has been judged scientifically suitable for publication and will be formally accepted for publication once it meets all outstanding technical requirements.

Kind regards,

Qunxi Gong

Academic Editor

PLOS ONE

Additional Editor Comments (optional):

Reviewers' comments:

Reviewer's Responses to Questions

**Comments to the Author**

1. If the authors have adequately addressed your comments raised in a previous round of review and you feel that this manuscript is now acceptable for publication, you may indicate that here to bypass the “Comments to the Author” section, enter your conflict of interest statement in the “Confidential to Editor” section, and submit your "Accept" recommendation.

Reviewer #3: All comments have been addressed

2. Is the manuscript technically sound, and do the data support the conclusions?

Reviewer #3: Yes

3. Has the statistical analysis been performed appropriately and rigorously? 

Reviewer #3: Yes

4. Have the authors made all data underlying the findings in their manuscript fully available?

Reviewer #3: Yes

5. Is the manuscript presented in an intelligible fashion and written in standard English?

Reviewer #3: Yes

6. Review Comments to the Author

Reviewer #3: This article is designed to analyze the logistics capacity, influencing factors and spatial spillover effect in Yangtze River Economic Belt. It' well revised. I tink it can be accepted now.

7. PLOS authors have the option to publish the peer review history of their article (what does this mean?). If published, this will include your full peer review and any attached files.

Reviewer #3: No

---

## [Editor Report · Acceptance letter]

8 May 2024

PONE-D-23-24153R2 

PLOS ONE

Dear Dr. Yinnan, 

I'm pleased to inform you that your manuscript has been deemed suitable for publication in PLOS ONE. Congratulations! Your manuscript is now being handed over to our production team.

Kind regards, 

on behalf of

Dr. Qunxi Gong 

Academic Editor

PLOS ONE